# Acute Appendicitis in COVID-19-Positive Children: Report of 4 Cases from an Endemic Area in Northeastern Romania

**DOI:** 10.3390/ijerph20010706

**Published:** 2022-12-30

**Authors:** Florin Filip, Monica Terteliu Baitan, Ramona Avramia, Roxana Filip

**Affiliations:** 1Faculty of Medicine and Biological Sciences, Stefan cel Mare University of Suceava, 720229 Suceava, Romania; 2Suceava Emergency County Hospital, 720224 Suceava, Romania; 3Synevo Laboratory, 720262 Suceava, Romania

**Keywords:** acute appendicitis, children, SARS-CoV-2, surgery, PCR test

## Abstract

Acute appendicitis (AA) is one of the most common surgical emergencies in children. Some reports have suggested that the COVID-19 pandemic was responsible for delays in the diagnostic and proper treatment of AA in pediatric patients. The aim of our study was to perform a retrospective study of cases of AA in children with SARS-CoV-2 infection treated in a highly endemic area for COVID-19 in Romania during a 2-year time interval. The SARS-CoV-2 infection had no unfavorable impact on children who presented with AA. Further data analysis should clarify the overall influence of COVID-19 on the management of surgical pediatric patients in such endemic areas.

## 1. Introduction

Severe acute respiratory syndrome coronavirus 2 (SARS-CoV-2) was identified for the first time in December 2019 in Wuhan, and since then it has spread very quickly throughout the world, finding victims among all categories of the population. From the end of December 2019 until December 2022, the number of COVID-19 cases exceeded 600 million, with over 6.4 million deaths worldwide. In Romania, the first case was reported on 26 February 2020, and on 22 March the Ministry of Health (MoH) confirmed the first death related to SARS-CoV-2 infection. The number of fatalities increased significantly, reaching the maximum on 8 December 2020 (213 registered deaths). As of the end of December 2022, the number of confirmed cases of COVID-19 in our country has exceeded 3.3 million, of which over 15% were children [1].

The worldwide spread of infection with the SARS-CoV-2 virus and the subsequent COVID-19 pandemic have had dramatic effects on many national health systems, both in Europe and worldwide [2,3]. It is generally accepted that children are less clinically involved than the adult population [3,4,5], developing mostly mild upper respiratory symptoms. However, many pediatric centers experienced a dramatic decrease in presentations in their pediatric emergency rooms (PERs) during the COVID- 19 pandemic.

Starting from the summer of 2021, in Romania, the number of children with COVID-19 has continued to increase, reaching a maximum of 559 new cases on 28 September 2021. According to the statistics, children present with milder forms, unlike adults, and the manifestations are usually asymptomatic. It is considered that these outcomes are due to the higher immunity level of children compared to the elderly, especially against common colds, but also to changes in the expression of the viral entry receptor ACE2 [6]. Later, however, the clinical manifestations in children acquired other characteristics, with the symptoms being similar to those of Kawasaki disease, including high fever, rash, conjunctivitis, gastrointestinal symptoms, encephalitis, coronary artery dilation, cardiovascular shock, and finally multiple organ failure [7].

Several reports have already estimated the impact of COVID-19 disease on pediatric admissions requiring surgery, such as acute appendicitis (AA). There has been a general perception that COVID-19 has delayed presentations for AA or was responsible for an increased incidence of complicated cases [8,9].

Not all studies, however, were able to confirm this perception [10,11]. Our paper relates to the management of COVID-19-positive pediatric patients with AA in a pediatric surgery department located in Suceava County, Northeastern Romania. The department is currently serving as a reference pediatric surgery unit for a population of over 800,000 (of which 150,000 are children). This area was affected by a high incidence rate of COVID-19 cases for almost two years, starting in March 2020, with over 58,000 illnesses reported per total number of inhabitants (105,000).

## 2. Materials and Methods

The methodology for this study was similar to other reports [6]. The hospital records of children with acute appendicitis (AA) reported during the COVID pandemic between 1 April 2020 and 31 March 2022 in our hospital were reviewed. This was the time interval with a significant incidence rate of COVID-19 disease in the surrounding geographical area. The patients’ demographics, age, and macroscopic and microscopic findings and the time interval between the onset of symptoms and PER presentation were included. Patients with non-confirmed appendicitis or with conservative treatment for AA were excluded. AA cases were classified as ‘complicated’ (abscess, peritonitis, perforated appendicitis) or non-complicated (lack of local or systemic complications) according to other papers [10,11]. The significant results of laboratory and ultrasound (US) studies performed at admission and during hospitalization were included if available. Screening for SARS-CoV-2 infection was performed at admission using either PCR or rapid antigen testing, according to the national COVID-19 protocol in use at that time. All pediatric patients with surgical emergencies were initially admitted to a buffer surgical area while waiting for the results of their PCR tests to become available. They were started on i.v. fluids, antibiotics (cefuroxime and metronidazole), and analgesics. For patients with positive results for SARS-CoV-2 infection, the surgery was performed in a COVID- designated operating room. After surgery, these patients were transferred to a special area in the hospital where only COVID-19-positive patients were admitted.

## 3. Results

A total of 4 AA pediatric patients who tested positive for SARS-CoV-2 infection were identified. The main data related to these patients are presented in Table 1.

There were no distant complications recorded in these patients. They were followed-up by the family practitioner with respect to their COVID-19 status, postoperative course, and return to regular activities.

## 4. Discussions

Acute appendicitis (AA) is the most common surgical emergency in children [12]. In the US, it accounts for nearly 70,000 cases in children per year, representing one-third of all admissions for abdominal pain in this age group. Over the years, AA has remained a major cause of surgical interventions in the pediatric population [13]. Children have significantly higher perforation rates compared with adults, especially those under 5 years of age [14]. There is also a peak incidence rate of cases between 12 and 18 years of age, with a very small number of cases in neonates [15].

The COVID-19 pandemic had a significant impact on the routine activity of pediatric surgery centers worldwide. Many pediatric facilities, both medical and surgical, have been reconverted into ‘COVID-19 wards’ and all planned admissions and procedures have been suspended. A reduction in the number of presentations in pediatric emergency rooms (PERs) has also been reported, especially in geographical areas with high incidence rates of SARS-CoV-2 infections such as in Northern Italy [9,11]. There has been a common perception that the presentation rate for abdominal pain, including cases of acute appendicitis (AA), in pediatric patients has decreased during the pandemic. Several reasons may justify a delay in presentation for AA patients (such as fear of getting exposed to SARS-CoV-2 infection in public places and a fear of presenting to the primary care physician). Snapiri et al. reported a higher rate of complications in AA patients during the COVID-19 pandemic [10]. La Pergola et al. did not confirm this assumption in their study, which compared the three previous years with data from 2020 at the peak of the COVID-19 pandemic in a highly endemic area of Northern Italy [11]. Additionally, La Pergola could not confirm a delay in the presentation of children with AA during pandemics when using a statistical analysis. They also found that there were no significant differences regarding the prevalence and the onset of symptoms of AA in children during the peak period of the pandemic when compared to the three previous years. Several other studies showed similar conclusions regarding pediatric patients who were also SARS-CoV-2-positive; these cases did not show delayed presentation or increased rates of complications [16,17]. Regarding the decreased rate of AA cases in adults during the COVID-19 pandemic, it may be explained by the successful treatment of mild cases at home [18].

The relationship between acute appendicitis (AA), SARS-CoV-2 infection, and MIS-C (Multisystem Inflammatory Syndrome in Children) has generated many discussions during the COVID-19 pandemic. In May 2020, the Centers for Disease Control and Prevention (CDC) issued a national health advisory regarding the cases that meet the criteria for MISC. This was defined as a dysregulated immune response with host tissue damage and hyperinflammation, resembling Kawasaki disease (KD), toxic shock syndrome (TSS), or macrophage activating syndrome. A possible connection between AA and KD was previously suggested, and it is well known that KD shares many common features with MISC, possibly related to appendicular artery vasculitis [19]. The median age of onset in patients with MISC was 8.3 years [20,21,22]. Most cases required hospitalization (80–88%) and intensive care admission (80%) for multiple organ dysfunction [12,20]. As part of the MIS- C definition, the patients were individuals of less than 21 years of age with fever for >24 h, laboratory evidence of inflammation, and recent SARS-CoV-2 infection or exposure within 4 weeks prior to the onset of symptoms. The organ systems commonly involved in MISC include the gastrointestinal (GI) (92%) and cardiovascular (80%) systems [20]. GI symptoms include abdominal pain, vomiting, diarrhea, and manifestations that can mimic acute appendicitis (AA), such as terminal ileitis and mesenteric adenitis [23]. MISC may represent a postinfectious hyperinflammatory complication of SARS-CoV-2 infection, since it develops later in the course of COVID-19. MIS-C patients are often PCR- negative but antibody-positive, suggesting a late manifestation of SARS-CoV-2 infection [24]. In order to support this hypothesis, Malhotra et al. [24] reported a peculiar pattern of admission in SARS-CoV-2-positive pediatric patients, with pneumonia cases representing early cases (first 6 weeks of the pandemic), followed 4 weeks later by cases of MIS-C and AA. Their study included a total of 48 SARS-CoV-2-positive children admitted to a tertiary care pediatric center during the peak interval of the COVID-19 pandemic in New Jersey (29 March to 26 July 2020). Pneumonia and MIS-C were the most frequent cases; a cluster of 10 (20.8% of total cases) SARS-CoV-2-positive patients were admitted with AA.

Tullie et al. reported that children with SARS-CoV-2 infection may present with clinical features suggestive of acute appendicitis (AA) or atypical AA as part of MISC [17]. In their series of pediatric cases from London, all patients were diagnosed with terminal ileitis and did not require surgery. Furthermore, Lishman et al. reported on 3 cases of children who were diagnosed with AA and who underwent surgery with confirmed appendicitis at pathology (2 of them with perforated appendicitis) [25]. However, all 3 patients were diagnosed with MISC after surgery based on extensive laboratory and cardiology (echocardiography) assessments. They required admission to the intensive care unit (ICU) and complex medical management including intravenous immunoglobulins (IVIG), inotropic support, a steroid i.v. pulse, and antibiotics. The patients were hospitalized for 7, 10, and 11 days, respectively, with the intermittent need for ventilation and mask-delivered O_2_ in 2 patients. Furthermore, deVos et al. suggested that AA may occur as a complication of SARS-CoV-2 infection through similar mechanisms as typical, non-COVID-related AA (inflammation associated with viral entry or reactive lymphoid hyperplasia causing luminal obstruction). The existence of a large number of angiotensin-converting enzyme 2 (ACE-2) receptors in the terminal ileum could facilitate the viral binding at this level, with consequent clinical and pathological features [26,27]. Furthermore, deVos also stated that if no fecaliths are found in children with AA, vasculitis or inflammation may represent the pathological mechanism of occurrence. In their opinion, there is a definite connection between AA, COVID-19, and MISC, and MISC should be considered in every patient with clinical AA who is also positive for SARS-CoV-2 at admission. This assumption has also been supported by other studies [24,25], which underlined the similitude of gastrointestinal symptoms found in children with MISC and AA, respectively.

Alotaibi et al. reported on 2 cases of AA in children who, based on clinical features and abdominal ultrasound (US) findings, were diagnosed with AA and underwent surgery [28]. The pathology was consistent with benign lymphoid hyperplasia in one case and acute appendicitis (but without perforation) in the second case. The postoperative course was difficult in both cases, with hypotension and ICU admission; additional laboratory findings and the patient history confirmed the diagnosis of MISC. The patients received appropriate treatment with antibiotics, IVIG, i.v. methylprednisolone, and prophylactic enoxaparin. They went home in a stable condition. Alotaibi et al. concluded that MISC can mimic AA and that both conditions can occur simultaneously. Using a multidisciplinary approach and obtaining relevant laboratory data and radiological imaging results can facilitate the early recognition and appropriate treatment of MISC, as well as the appropriate timing of surgery for AA in MISC cases. This approach allows the early recognition of MISC in patients presenting with clinical symptoms suggestive of acute abdomen, including AA. It is cautious because it can avoid unnecessary surgical interventions, which is advantageous considering the high risk of morbidity in these patients [29]. Moreover, patients with uncomplicated AA can be safely treated conservatively [12,30]. Patients with AA and MISC are likely to require surgical procedures due to the severe presentation of MISC, which may lead to a clinical decision for an appendectomy. Anderson et al. reported the successful management of a patient with AA in whom concomitant MISC was recognized and treated with IVIG, antibiotics, and inotropic support until the patient’s condition became stable [31]. Surgery was decided on after identifying a fecalith in the appendiceal lumen, and a perforated appendicitis was found during surgery. After surgery, the patient received an interleukin- 1 receptor antagonist (Anakinra) instead of corticosteroids.

Suceava County Hospital, where this study was performed, is a large, 1400-bed unit, covering the medical assistance for more than 800,000 people, of which 150,000 are children. Starting at the end of March 2020, Suceava County was placed under epidemiological quarantine until late May 2020. The hospital was designated as a ‘COVID-19 hospital’ for various time intervals during the COVID-19 pandemic. As a result, many emergency or scheduled presentations had to be canceled until 1 June 2020, when the hospital again started regularly functioning. This was also true for the pediatric surgery department. Until the end of May 2020, all surgical cases in pediatric patients had to be directed to hospitals in close proximity for appropriate treatment. The regular activity started again on 1 June 2020; between this date and 31 March 2022 (the beginning and end points of our study, respectively), a total of 1598 patients were discharged from our department. This number represented a significant decrease compared to a similar time interval 2 years before (1 June 2018–31 March 2020), with 2334 patients discharged (a decrease of 31.54%). Additionally, the number of children with acute appendicitis (AA) discharged during the 2-time intervals decreased from 324 cases (1 June 2018–31 March 2020) to 172 (1 June 2020–31 March 2022), a decrease of 46.3%. We consider these changes to be mostly related to COVID-19 pandemic, similar to other reports worldwide. Of the 172 cases of AA treated between 1 June 2018 and 31 March 2022, 4 (2.32%) were also diagnosed with SARS-CoV-2 infection and were included in this study.

## 5. Conclusions

We have reported on a total of 4 (four) cases of acute appendicitis (AA) in SARS-CoV-2-positive patients from a pediatric surgical department in Northeastern Romania, which experienced a large number of COVID-19 cases. No patient had respiratory symptoms at presentation, and only 1 (one) of them had a fever, which was not related to the abdominal disease. The management of these patients was adapted to the COVID-19 pandemic protocols, including a 12–24-h delay before the result of the PCR test became available. This did not generate any immediate unfavorable outcomes. There was no patient with complicated AA at the time of surgery or clinical or laboratory data to support the existence of MISC with pseudo-appendicitis as a clinical picture. Only one patient required ICU admission, but none of them required inotropic support or systemic anti-inflammatory therapy postappendectomy. The pathology was consistent with acute appendicitis in all 4 cases. There were no long-term complications recorded in these patients. Although the number of cases was small (but comparable with other studies), we considered these cases to be AA in SARS-CoV-2-positive patients and not gastrointestinal manifestations of MISC.

## Figures and Tables

**Table 1 ijerph-20-00706-t001:** The clinical and laboratory findings for children with AA and SARS-CoV-2.

CASE	1	2	3	4
Age	8	13	15	8
Sex	F	M	F	F
SARS-CoV-2	+	+	+	+
Diagnostics	acute phlegmonous appendicitis	acute phlegmonous appendicitis	acute phlegmonous appendicitis	acute phlegmonous appendicitis
Signs and symptoms				
Duration of symptoms prior to surgery, days	1	4	1	2
Fever	-	+	-	-
Respiratory symptoms	-	-	-	-
Abdominal pain	+	+	+	+
Vomiting	-	-	+	+
Diarrhea	-	+	-	-
Nausea			+	+
Conjunctival injection	-	-	-	-
Shock	-	-	-	-
Laboratory findings				
CRP, mg/dl	2.29	7.73	9.2	-
Total WBC, ×10^3^/µL	10.21	17.06	13.18	15.80
Lymphocytes, ×10^3^/µL	1.08; 10.67% of total WBC	2.41; 14.1% of total WBC	0.55; 4% of total WBC	1.39; 8.8% of total WBC
Monocytes, ×10^3^/µL	1.76; 17.2% of total WBC	1.71; 10% of total WBC	0.33; 2.4% of total WBC	0.47; 3% of total WBC
Neutrophils, ×10^3^/µL	6.97; 67.9% of total WBC	12.54; 73.5% of total WBC	12.11; 92.7% of total WBC	13.73; 86.9% of total WBC
Creatinine, mg/dl	0.45	0.75	0.6	0.45
ALAT, U/L	12	31	12	18
AST, U/L	25	35	20	25
Glucose, mg/dl	96	92	97	96
Management				
Intensive care, hours	-	24	-	-
Analgesics	+	+	+	+
Antibiotics	+	+	+	+
Respiratory support	-	-	-	-
Duration of hospitalization, days	3	3	14	3
Recommendations	proper isolation precautions and 14 days of isolation with SARS-CoV-2 test at the end of the isolation interval	regular isolation instructions according to the COVID-19 protocol	no Isolation after discharge	regular isolation instructions according to the COVID-19 protocol
Imaging				
Abdominal US	no significant changes, but the clinical picture highly significant for AA	dilated, 18 mm, appendix lumen; no free peritoneal fluid or lymphadenopathy	dilated, 9 mm, appendicular lumen; infiltrated peri appendiceal fat with minimal free peritoneal fluid; no lymphadenopathy	dilated, 10 mm, uncompressible appendicular lumen; infiltrated periappendiceal fat; no peritoneal free fluid or lymphadenopathy

Normal ranges: WBC values: 5–10 × 10^3^/µL; normal lymphocytes values: 1.3–4; 21–40% of total WBCs; normal neutrophil values: 2–7.5; 40–75% of total WBCs; creatinine: 0.57–0.8; CRP: <0.5; ALAT: 9–24; AST: 18–36; glucose: 60–99—normal; 100–125—modified basal glucose; >126: diabetes mellitus (DM).

## Data Availability

Not applicable.

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
