# Peer review of "Acute Appendicitis in COVID-19-Positive Children: Report of 4 Cases from an Endemic Area in Northeastern Romania"

_ijerph, 2022, doi:10.3390/ijerph20010706_

Round 1

Reviewer 1 Report

Filip et al. described 4 cases of acute appendicitis diagnosed during the COVID-19 pandemic. Despite some issues, this study is interesting.

INTRODUCTION

What does the phrase  “a high incidence of COVID- 19 cases" mean?

It would be useful to add more information about the involvement of the gastrointestinal tract during SARS-CoV-2 infection, for example: https://doi.org/10.5114/ceji.2022.116368

RESULTS

It seems that it will be clearer to put information about patients (age, duration of symptoms, results of laboratory tests) in a table.

DISSCUSION

Line 114-117: What does "during the New Jersey COVID-19 pandemic" mean?

Line 123: The authors mention only one reason for the delay in diagnosis. What other reasons could there be?

Line 136-138: Do the given numbers of patient admissions refer to the pandemic period? Do you know how many AAs were diagnosed during the observation period? What % were patients with COVID-19?

CONCLUSSION

Based on 4 cases, it is difficult to draw conclusions, I suggest taking this into account.

Author Response

19.12.2022

MDPI AG, St. Alban-Anlage 66

4052 Basel, Switzerland

Tel.: +41 61 683 77 34

Journal: International Journal of Environmental Research and Public Health

Special Issue: Pediatric and Primary Health Care Services in the 21st Century

Re:        Manuscript ID ijerph-2086807

Esteemed Reviewer:

We greatly appreciate the opportunity to have our work reviewed by you. The following phrases include your comments and our responses.

Filip et al. described 4 cases of acute appendicitis diagnosed during the COVID-19 pandemic. Despite some issues, this study is interesting.

Thank you for the review and for the suggestions offered.

  1. What does the phrase “a high incidence of COVID- 19 cases" mean?

Response: In our country, there were more than 58,000 people infected with COVID-19, compared to the total number of inhabitants (105,000). We added this information in lines 51-52. The situation form our country was mentioned in lines 21-26. Additional information regarding our hospital are presented in lines 187-205.

  1. It would be useful to add more information about the involvement of the gastrointestinal tract during SARS-CoV-2 infection, for example: https://doi.org/10.5114/ceji.2022.116368

Response: Thank you for the observations. We added the information in discussion part, lines 129-141. The reference you suggested was added at no. 31

  1. It seems that it will be clearer to put information about patients (age, duration of symptoms, results of laboratory tests) in a table

Response: We did it, thank you for your suggestion. Table 1, line 76.

  1. Line 114-117: What does "during the New Jersey COVID-19 pandemic" mean?

Response: We compare our findings with those of Malhotra el al. who explain the situation of positive children admitted to a tertiary care pediatric center during the peak interval of COVID - 19 pandemic in New Jersey (March 29 to July 26, 2020). Pneumonia and MIS-C were the most frequent cases; a cluster of 10 (20.8% of total cases) SARS- CoV- 2 positive patients were admitted with AA. We noted this information on lines 138-141.

  1. Line 123: The authors mention only one reason for the delay in diagnosis. What other reasons could there be?

Response: Besides the fact that, during the pandemic, the hospital was considered one of the first line, the patients' fear of being infected once they arrived at the emergency unit was another factor (lines 99-101).

  1. Line 136-138: Do the given numbers of patient admissions refer to the pandemic period? Do you know how many AAs were diagnosed during the observation period? What % were patients with COVID-19?

Response: Thank you for the suggestion. Until the end of May 2020, all surgical cases in pediatric patients had to be directed to hospitals in close proximity for appropriate treatment. The regular activity started again on June 1st, 2020; between this date and March 31st, 2022 (beginning and end points of our study, respectively) a number of 1.598 patients were discharged from our department. This number represented a significant decrease compared to a similar time interval 2 years before (June 1st, 2018- March 31st, 2020), with 2.334 patients discharged (a decrease of 31.54%). Also, the number of children with acute appendicitis (AA) discharged during the 2-time intervals de-creased from 324 cases (June 1st, 2018- March 31st, 2020) to 172 June 1st, 2020- March 31st, 2022), a decrease of 46.3%. We consider these changes to be mostly related to COVID- 19 pandemic, similar to other reports worldwide. Of the 172 cases of AA treated between June 1st, 2018, and March 31st, 2022, 4 (2.32%) were also diagnosed with SARS- CoV- 2 infection and were included in this study. We presented the information in lines 193-205.

  1. Based on 4 cases, it is difficult to draw conclusions, I suggest taking this into account.

Response: It is true, but we also wanted to expose the situation encountered in a developing country, with major implications due to COVID-19. Also, we consider that the development of the discussion section, by referring to other similar or comparative situations, improves the paper and can be useful to the people directly involved.

Best Regards,

MD Roxana Filip, Ph.D.

Corresponding Author
[email protected]

Reviewer 2 Report

Dear Authors,

The paper deals with a topical issue and is new, then I will comment on some aspects that need to be improved.

The abstract should have a clearer structure in which the objective is discussed (v.g The aim of this study is……), as well as the results and conclusions.

In the summary methodology, it must be indicated that there are 4 cases that make up the study.

This phrase “In our opinion, the SARS –CoV–2 infection had no unfavorable impact on children who presented with AA” must be qualified, it is not about giving an opinion, but rather arguing this with scientific evidence, if any, or based on to the results obtained in the study.

The introduction should be expanded, justifying the reasons for this investigation.

Regarding the study methodology, using only four cases seems to me little to be able to draw conclusions on this topic. If more cases have been carried out, it would be convenient if they were provided in order to enrich the study.

It would be necessary to carry out statistical analyzes comparing the different cases with each other, selecting a series of variables that can be predictors to associate coronavirus with appendicitis.

On the other hand, the discussion should be supported by more references on this subject, since many studies on this virus have proliferated almost three years ago. Or justify the lack of studies that link testing positive for coronavirus with appendicitis.

Author Response

19.12.2022

MDPI AG, St. Alban-Anlage 66

4052 Basel, Switzerland

Tel.: +41 61 683 77 34

Journal: International Journal of Environmental Research and Public Health

Special Issue: Pediatric and Primary Health Care Services in the 21st Century

Re:      Manuscript ID ijerph-2086807

Esteemed Reviewer:

We greatly appreciate the opportunity to have our work reviewed by you. The following phrases include your comments and our responses.

The paper deals with a topical issue and is new, then I will comment on some aspects that need to be improved.

Response: Thank you for comments. We appreciate your effort and time.

  1. The abstract should have a clearer structure in which the objective is discussed (e.g The aim of this study is……), as well as the results and conclusions.

Response: We modified, thank you (lines 10-11). We added information in results, discussions and conclusions part (lines 75-84, 86-206, 216-221).

  1. In the summary methodology, it must be indicated that there are 4 cases that make up the study.

Response: We added, thank you. Line 75.

  1. This phrase “In our opinion, the SARS –CoV–2 infection had no unfavorable impact on children who presented with AA” must be qualified, it is not about giving an opinion, but rather arguing this with scientific evidence, if any, or based on to the results obtained in the study.

Response: We changed it, thank you. We added information in results and discussions sections (lines 74-83, 85-205).

  1. The introduction should be expanded, justifying the reasons for this investigation.

Response: We added information which can be find in lines 17-26 and 33-41).

  1. Regarding the study methodology, using only four cases seems to me little to be able to draw conclusions on this topic. If more cases have been carried out, it would be convenient if they were provided in order to enrich the study.

Response: The presented cases are the only ones with a confirmed positive test and AA. To support the information presented, I have completed the discussion part with findings from the relevant literature (Discussion part – lines 85-205). We also added additional information about the cases in our hospital (lines 195-205).

  1. It would be necessary to carry out statistical analyzes comparing the different cases with each other, selecting a series of variables that can be predictors to associate coronavirus with appendicitis.

Response: According to the number of cases presented (4), statistical analysis with significant results cannot be performed. Even the student test, which involves the lowest number of entries, can be applied from 7 cases upwards. However, we have included table 1 and noted the information so that it can be easily viewed and discussed.

  1. On the other hand, the discussion should be supported by more references on this subject, since many studies on this virus have proliferated almost three years ago. Or justify the lack of studies that link testing positive for coronavirus with appendicitis.

Best Regards,

MD Roxana Filip, Ph.D.

Corresponding Author
[email protected]

Reviewer 3 Report

Altough the topic is interesting with only 4 cases you cannot perform a study and to draw valid conclusions at national level.

The introduction section  is too short.

The material and methods section is not properly described.

According to the results section, this is definitely not a study, it is only a series of cases.

Author Response

19.12.2022

MDPI AG, St. Alban-Anlage 66

4052 Basel, Switzerland

Tel.: +41 61 683 77 34

Journal: International Journal of Environmental Research and Public Health

Special Issue: Pediatric and Primary Health Care Services in the 21st Century

Re:        Manuscript ID ijerph-2086807

Esteemed Reviewer:

We greatly appreciate the opportunity to have our work reviewed by you. The following phrases include your comments and our responses.

Although the topic is interesting with only 4 cases you cannot perform a study and to draw valid conclusions at national level.

Response: Thank you for the suggestions. It is true, but we also wanted to expose the situation encountered in a developing country, with major implications due to COVID-19. Also, we consider that the development of the discussion section, by referring to other similar or comparative situations, improves the paper and can be useful to the people directly involved.

  1. The introduction section is too short.

Response: We added new information (lines 17-26 and 33-41).

  1. The material and methods section is not properly described.

Response: We organized the information into Table (Table 1). On the other hand, we used the template intended for case reports and described as it is mentioned in other works of this type, published by the IJERPH journal.

  1. According to the results section, this is definitely not a study, it is only a series of cases.

Response: The aim was not to write a study, but to present a series of cases. Therefore, the manuscript was adapted according to the template and placed in the area of case reports. We have also checked other studies that present fewer cases and we can provide you with a series of links where they can be found:

- one case presentation: https://www.mdpi.com/1660-4601/18/15/8057; https://www.mdpi.com/1660-4601/19/24/16652; https://www.mdpi.com/1660-4601/18/11/5882;

- two cases presentation: https://www.mdpi.com/1660-4601/19/18/11437; https://www.mdpi.com/1660-4601/19/5/2761:

- three cases presentation: https://www.mdpi.com/1660-4601/19/21/13798.

Furthermore, according to IJERPH Journal, “Case reports present detailed information on the symptoms, signs, diagnosis, treatment (including all types of interventions), and outcomes of an individual patient.”

                        Best Regards,

MD Roxana Filip, Ph.D.

Corresponding Author
[email protected]

Round 2

Reviewer 2 Report

Dear authors, 

Thank you for your contribution, the study has been improved, 

kind regards,